# Deep Learning Models for Classification of Dental Diseases Using Orthopantomography X-ray OPG Images

**DOI:** 10.3390/s22197370

**Published:** 2022-09-28

**Authors:** Yassir Edrees Almalki, Amsa Imam Din, Muhammad Ramzan, Muhammad Irfan, Khalid Mahmood Aamir, Abdullah Almalki, Saud Alotaibi, Ghada Alaglan, Hassan A Alshamrani, Saifur Rahman

**Affiliations:** 1Division of Radiology, Department of Internal Medicine, Medical College, Najran University, Najran 61441, Saudi Arabia; 2Department of Computer Science and Information Technology, University of Sargodha, Sargodha 40100, Pakistan; 3Electrical Engineering Department, College of Engineering, Najran University Saudi Arabia, Najran 61441, Saudi Arabia; 4Department of Preventive Dental Sciences, College of Dentistry, Majmaah University, Al-Majmaah 11952, Saudi Arabia; 5Department of Orthodontics and Pediatric Dentistry, College of Dentistry, Qassim University, Buraidah 51452, Saudi Arabia; 6Radiological Sciences Department, College of Applied Medical Sciences, Najran University, Najran 61441, Saudi Arabia

**Keywords:** BDR, deep learning, OPG, YOLO, dentistry, annotation, augmentation, medical imaging

## Abstract

The teeth are the most challenging material to work with in the human body. Existing methods for detecting teeth problems are characterised by low efficiency, the complexity of the experiential operation, and a higher level of user intervention. Older oral disease detection approaches were manual, time-consuming, and required a dentist to examine and evaluate the disease. To address these concerns, we propose a novel approach for detecting and classifying the four most common teeth problems: cavities, root canals, dental crowns, and broken-down root canals, based on the deep learning model. In this study, we apply the YOLOv3 deep learning model to develop an automated tool capable of diagnosing and classifying dental abnormalities, such as dental panoramic X-ray images (OPG). Due to the lack of dental disease datasets, we created the Dental X-rays dataset to detect and classify these diseases. The size of datasets used after augmentation was 1200 images. The dataset comprises dental panoramic images with dental disorders such as cavities, root canals, BDR, dental crowns, and so on. The dataset was divided into 70% training and 30% testing images. The trained model YOLOv3 was evaluated on test images after training. The experiments demonstrated that the proposed model achieved 99.33% accuracy and performed better than the existing state-of-the-art models in terms of accuracy and universality if we used our datasets on other models.

## 1. Introduction

Dental informatics is a new field in dentistry that helps and improves dental practice diagnosis procedures, saves time, and reduces stress in people’s daily lives [1]. The main areas of dentistry are restorative dentistry, endodontics, orthodontics, dental surgery, and periodontology. Restorative dentistry refers to any dental procedure that repairs or replaces a tooth. Dental work and root canals are examples of restorative procedures. Endodontics is the branch of dentistry that deals with the dental pulp and the cells surrounding tooth roots. Orthodontics is a branch of dentistry that deals with tooth abnormalities and how to correct them. Dental surgery encompasses a wide range of medical treatments that involve the intentional modification of dentition, that is, surgery on the teeth, jaw bones, and gums. Periodontology is a branch of dentistry that deals with diseases of the teeth’s supporting and investing tissues, such as cementum, periodontal ligaments, gums, and alveolar bone [2]. The following are the most common dental illnesses: cavities are permanently damaged areas of the tooth’s hard surface that develop into small gaps or holes. Dental crowns are caps that go over damaged teeth. Crowns are used to protect, cover, and restore the form of teeth when fillings fail to solve the problem. A root canal is a treatment that removes tooth infection by diagnosing and treating a tooth’s diseased pulp.

Dentists are now responsible for diagnosing tooth problems. They can detect potential dental problems by inspecting and gently moving the teeth. There has been little progress in the automatic detection of dental problems. For disease classification and detection, manual analysis of teeth problems necessitates time and expertise. In manual analysis, human error can lead to incorrect predictions. The automatic dental problem detection and classification system will aid in early disease diagnosis and may prevent tooth loss. It will aid in eliminating manual clinical examination, which is time-consuming, tedious, and labour-intensive. In the past, medical imaging technologies such as CT and X-rays have greatly aided in treating and diagnosing various diseases [2].

An X-ray generator can generate radiographic X-rays passing through the mouth as tissues absorb radiation. The procedure known as projective-radiography produces two-dimensional (2D) images of the human body’s internal architecture [3]. The introduction of high-resolution biosensors and sensor images has resulted in massive amounts of data that can be analysed using computer programmes to assist dental professionals in making prevention, treatment planning, and diagnosis decisions [4]. Dental radiographs are classified as intraoral (the film is placed within the buccal space) or extraoral (the patient is placed between the X-ray source and the radiographic film). A panoramic dental radiograph displays the entire area of the mouth, including all of the teeth. Preferred medical image-processing solutions, such as CNN, have been used in various clinical settings [5,6]. A convolutional neural network has been developed as a strong machine learning technique, capable of tackling tasks such as image identification, segmentation, and classification with high accuracy. Deep CNN techniques were developed to detect deterioration, periapical periodontitis, and periodontal diseases of mild, moderate, and severe severity on clinical dental periapical radiographs. The CNN model investigated automatic periapical radiograph feature recognition, segmentation, and quantification. In every metric, the U-Net design and its version outscored Xnet and SegNet [7]. A deep learning algorithm was developed to detect and locate dental lesions in infrared transillumination images [8] and ultrasound image detection of hepatocellular carcinoma regions [9]. CNNs have been used in dentistry [10], caries detection [11], and apical lesions detection [12] to detect PBL. CNNs can also detect, classify, and segment different structures [13]. In this work, we describe a deep learning-based solution for supporting dentists in correctly identifying patients’ dental problems using panoramic dental X-ray pictures.

The proposed oral health care system can be used as a clinical assistant for discovering dental problems. It is a cost-effective, robust, and efficient system that will significantly contribute to oral healthcare. Manual analysis of teeth problems requires time and expertise for disease classification and detection. Furthermore, in manual analysis, there is a chance of wrong predictions due to human error/misunderstanding. However, the automated dental problem detection and classification system will help in early diagnosis and may prevent significant problems such as tooth loss. It will also help eliminate manual, time-consuming, tedious, and extensive examinations. For these reasons, we propose a deep learning model, YOLOv3, which we trained and tested on our data set (no public teeth data set was available). The process of dataset collection and making our own custom dataset is a prominent strength of the proposed tooth abnormality classifier system. The dataset contains 800 panoramic X-rays of teeth with various tooth diseases. After image enhancement, also known as augmentation, various image variations such as horizontal flip, vertical flip, shear range, and zoom range meant that the number of images is increased to 1200. The limitation of this method is that the sample size for only four types of disorders restricted hospital data and was not representative of the overall population. The proposed methodology obtained the highest accuracy, 99.33%. The following are the primary contributions of the proposed work:The first step was to prepare and enhance the dataset. A limited dataset with 116 images is available on Kaggle for dental disease detection. For the study of dental diseases, a unique dataset was created. An expert BDS doctor performed data labelling for this domain’s data set processing. The datasets contain four different types of classes. During this phase, an OPG dataset of various patients was collected from three clinics. The dataset contains 800 panoramic X-rays of teeth with various tooth diseases. We performed the augmentation process, which includes various image variations such as horizontal flip, vertical flip, shear range, and zoom range. Finally, image annotation was performed with the LabelImg tool, which generated an annotated file in.txt format for each image.The second phase entailed training the dataset with the deep learning model YOLO (you only look once) version 3. We utilised this model by using sets of 800 images, 1000 images, and 1200 images. The best results were achieved after augmentation to 1200 images.The remainder of the paper is structured as follows. Section 2 examines previous research on dental diseases, Section 3 covers the study methodology and architecture of the proposed strategy, and Section 4 describes the experimental setup and outcomes. The final portion reviews the findings and prospects for further study.

## 2. Related Work

Dental informatics has developed many teeth segmentation strategies using contrary radiography images such as panoramic imaging, bitewing, and periapical images [14]. In [15], the authors compare ten segmentation algorithms used in dental imaging. To achieve instance segmentation, the authors of the work presented in [16] offer a method for tooth instance segmentation in panoramic pictures utilising a “mask region-based convolution neural network (CNN)”. The bounding box is drawn around the teeth, and the teeth are segmented in the last phase. The disadvantage of this procedure is that it concentrates solely on tooth detection, ignoring other issues like dentures and areas from where teeth are missing. DeNTNet, a deep neural transfer network that identifies the tooth disease named periodontal bone loss, or PBL, using panoramic dental radiographs, was proposed in [17]. Several convolutional neural networks were trained as part of the detecting procedure. To begin, training was done to extract teeth from the region of targeted interest with the help of a segmentation network; after that, forecasting periodontal bone loss lesions was done via a segmentation network. DeNTNet has the advantage of providing information about corresponding teeth numbers impacted by PBL according to the notation of the dental federation. Still, the disadvantage of this approach was detecting only this dental disease. For identifying and recognising a certain person after death, the authors of [18] offer a method by comparing a database of dental radiographs based on a set of unique traits with the postmortem of dental radiograph. For a modest image database, the results of this technique are promising, but human intervention is essential for configuring the algorithm settings and repairing any errors that may emerge. Zhang et al. [19] employed deep learning using the methodology for recognising and identifying teeth in dental periapical radiographs. Tooth loss, rotting teeth, and filled teeth, typical dental disorders in patients, were identified by combining fully convolutional neural networks based on the region (R-FCN) and faster R-CNN. The authors of [20] presented another technique that involves feeding and building features into a perceptron neural network which would be multi-layer with the goal of dental caries detection. The authors of [21] use two models of multi-sized CNN to recognise and classify teeth present in the dental panoramic radiographs to automatically structure the filing of dental charts. The testing data set achieved good object detection network accuracy by using a four-fold cross-validation procedure.

However, these approaches are simply concerned with teeth detection, not with the issues of classifying each of their faces. In their study, Fukuda et al. use CNNs to detect vertical root fracture (VRF) in panoramic radiographs [22]. The CNN used was created with Detect Net five-fold cross-validation, and DIGITS version 5.0 was employed to improve model dependability.

Lee et al. [23] used a deep convolutional neural network to analyse the radiographs and calculate each tooth’s radiographic bone loss (RBL). The RBL%, staging, and presumptive diagnosis of the revised CNN periodontitis classification were compared to the findings of independent examiners

The neural network achieved an accuracy rate of 85%. Neural networks may be valuable for evaluating radiographic bone loss and generating image-based periodontal diagnostics [24].

Artificial intelligence may be used to reveal dental restorations. AI may be applied in restorative dentistry to identify and categorise dental restorations, as demonstrated in a study from 2020 by Abdalla-Aslan R et al. On 83 panoramic photos, the algorithms utilised in their research identified 93.6% of dental restorations. In addition, restorations were categorised into 11 groups using the distribution and form of the grey values [25].

Convolutional neural networks were used by Krois et al. to analyse panoramic radiographs and identify periodontal bone loss as a proportion of the length of the tooth root. The outcomes were contrasted with the findings of six skilled dentists’ measurements. When it came to identifying periodontal bone loss, the CNN performed better than dentists (83%) in terms of accuracy and reliability (80%) [26].

Chang et al. [27] used panoramic images and convolutional neural networks to detect periodontal bone level (PBL), cementoenamel junction level (CEJL), and teeth in order to produce a periodontitis stage diagnosis.

The bone loss percentage was estimated and categorised by an automated system in the study by Jun-Young Cha et al. This technique can be used to gauge how severe peri-implantitis is [28].

Convolutional neural networks were utilised in a study by Byung Su Kim et al. to forecast if third molar extraction could result in inferior alveolar nerve paresthesia. Lower third molar extraction is one of the most common dental surgical procedures. Following the removal of a mandible wisdom tooth, the nerve may experience paresthesia. Prior to the extraction, panoramic pictures were taken, and CNN used the relationship between the nerve canal and tooth roots to forecast the likelihood of nerve paresthesia. Future research is required since, according to scientists, using two-dimensional pictures as panoramic radiography may produce more false positive and false negative outcomes [29]. All existing techniques for the dental diseases are discussed in Table 1.

## 3. Proposed Methodology

This research proposes a deep neural network model YOLOv3 for dental issue classification. The proposed method uses the orthopantomography panoramic teeth X-ray dataset to identify different dental disorders. The proposed architecture’s work is described as follows.

### The Proposed Architecture

This section explains the flow of our proposed system. The first step in our research was to collect the dataset we needed. After obtaining the dataset, it went through the preprocessing phase. It was further filtered where augmented and annotated, and images containing useless data are excluded. After preprocessing, it was divided into training and test segments and compressed before being uploaded to the drive. The Google Collab notebook was set to GPU, and the Dark Net repository was imported. Google Drive was mounted with Collab. A special folder called ‘YOLOv3′ was created in Google Drive, containing all necessary data. Other files that are required for coding were created. These files were called train.txt and were generated by Python code for a list of training images. For the test images, the same file was created. Another file was created with the names of all teeth classes. Another file was created that contains the links to all the files required for our work. All of these files, as well as the processed dataset, were imported into Google Collab. The training phase started, containing 8000 iterations. After 1000 iterations, a weight file was saved in Google Drive’s backup folder. The best result in the weight file and the final weight files were also stored for later evaluation. Weights were obtained and evaluated using performance evaluation metrics. These are the mean average precision (mAP), F1-score, recall, and IOU parameters (intersection over union). The proposed architecture’s workflow is illustrated in Figure 1.

## 4. Materials and Methods

In this section, all the steps are explained in detail. The first step is dataset collection, the second step is image augmentation, the third step is image annotation with the help of a labelling tool and lastly, training of the proposed mode with the help of YOLOv3.

### 4.1. Dataset Collection

The dataset contains OPG dental panoramic X-rays collected from clinics. Some OPGs were taken with a DSLR camera, whereas others were obtained from clinics in soft form. All the images are high resolution. The custom dataset contains 1200 images of patients ranging in age and teeth problems. The Figure 2 show our custom dataset:

### 4.2. Image Augmentation

Augmentation is the process of increasing the size of a dataset to meet our needs. We collected approximately 800 images from various clinics. Following that, we used some augmentation functions to increase the number of images. We ended up with 1200 images in total. Custom dataset after augmentation are shown in Figure 3. We used the image data generator function for augmentation with the following parameters:Rotation rangeZoom rangeShear rangeHorizontal flipVertical flip

**Figure 3 sensors-22-07370-f003:**
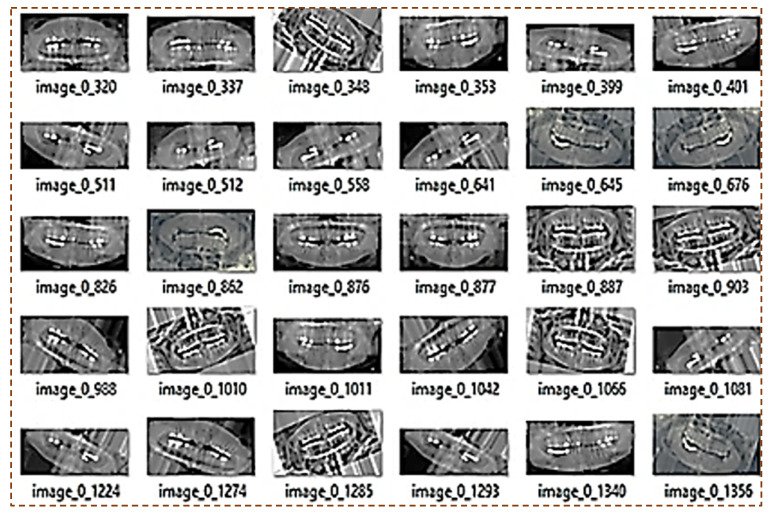
Custom dataset after augmentation.

### 4.3. Image Annotation

One of the significant concepts in deep learning is image annotation, by which we labelled specific parts of images for training the model. The YOLOv3 model, which accepts annotation files in “txt” format, was used. The image and the annotation file must be saved in the same archive. Each row in the annotation file represents a single bounding box in the image and includes the information listed below.
“<Object-class-id> <center-x> <center-y> <width> <height>”

These parameters are explained as follows:Objects-Class-Id: an int-type value that shows a class. The range of object-class-id is 0 to the number of classes−1. We have a total of 4 classes, so the range of object-class-id is 0 to 3.Center-x & Center-y: the center-x and center-y are the “center coordinates” of the bounding box.Width and Height: these are the dimensions of the bounding box.4.4 LabelImg.

LabelImg is a free open-source tool for graphically labelling images. It is written in Python and has a QT-based graphical user interface. It is a quick and simple method for labelling a few hundred images for object detection. Labeling process with LabelImg is shown in Figure 4.

Annotations were stored in the PASCAL VOC format as XML files which ImageNet uses [30]. It also supports the YOLO and Create ML formats. The annotation file for YOLO in .txt format is shown in the Figure 5:

### 4.4. Yolo (You Only Look Once)

J. Redmon et al. were the first to use YOLO in 2015. YOLOv1, YOLOv2, and YOLOv3 are three types of object detectors and classifiers. YOLOv1 has a problem with localisation and recall values that are not as good as the other models. YOLOv2 is entirely based on the CPU and requires more time to train. YOLOv3 was implemented in May 2018 [31]. It is a cutting-edge detector that produces better results in terms of speed and accuracy and is intended for GPU use. We used YOLOv3, by implementing its dark net framework, because of its performance and better results. YOLOv3′s detection procedure differs from that of other models. It only takes an image and sends it to the network once for further processing. The image is divided into S × S grid cells. Every grid cell’s responsibility is to predict the object centred in that grid cell and its bounding boxes, using their confidence scores. Confidence scores look at how likely objects are to exist and how confident they are in their prediction. In each grid cell, the conditional class probabilities should be observed. It is made up of 24 Conv layers and two fully connected layers. Some layers assemble the initial module, beginning with 1 × 1 reduction layers and progressing to 3 × 3 convolutional layers [32]. We used the dark net framework to put YOLO together. There are 53 network layers in the dark net. The residual block is a new type of block used by Darknet-53. Deep neural networks are difficult to train. As the network’s depth increases, the network’s accuracy can become saturated, resulting in increasing training error. The residual block was created to alleviate this issue. In terms of architecture, the insertion of a skip connection distinguishes the conventional convolution block from the residual block. The skip connection moves the input to higher layers. To avoid diminishing gradient issues as assistant activations spread across deeper layers, the Reset introduced the concept of skip connection. Overlapping layers will not impact network performance thanks to ResNet’s residual blocks. The structure of the dark net is elaborated in the following Table 2:

### 4.5. Custom Configuration File

Training requires the configuration file. Convolutional layers, YOLO masks, and other parameters are included. It was obtained from a darknet website. The configuration file must meet our specifications.

Batch: the batch parameter specifies the batch size used in training. There are 1200 images in our dataset. During the training process, the neural network is iteratively updated. It is impossible to update weight in training using all images at once. As a result, the set of images is used in a single iteration known as batch size.Subdivisions: if the batch size is set to 64 and the GPU does not have enough memory for training, Darknet allows you to set off the subdivision’s parameters by multiples of two, such as 2, 4, 8, 16, 32 until the training process is complete.Width, Height, Channel: the input image is resized in width and height before training. The resized image’s default size is 416 × 416. The results will be better if we increase the size, but training will take longer. The channel specifies whether the input image for training is grayscale or RGB.Learning rate, Steps: the learning rate governs how quickly the model adapts to the problem. The learning rate ranges from 0.01 to 0.0001. The learning rate is high at first, but it should gradually decrease. Decreasing of learning rate is specified in steps.Max batches, Classes: this defines the number of iterations that will be done on the training dataset. The number of iterations is set by the following formula 2000 * Classes. In our case, we have four classes, so we should train the dataset in at least 8000 batches.Filters: the filters of convolutional layers following the YOLO mask are set according to formula 3 * (5 + Classes). In our case, we have 30 filters.We made some changes to the configuration file parameters shown in Table 3.

## 5. Results and Discussion

In this section, we will evaluate the performance of our intended model, YOLOv3. The model will be tested on a panoramic X-ray dataset, as described in the document’s first section. Now, however, we will only discuss the experimental results obtained after applying the YOLOv3 model to a dataset of tooth OPGs. The model’s setup is explained in this section. The dataset was annotated on Windows using Python and the LabelImg module. We used Google Collab with Python and Open CV for testing and training. The specification of the Google Collab in Table 4 is given below:

We divided the dataset into sections of 70 and 30%. To begin, 70% of the dataset was used for training, while the other 30% of the dataset was used for testing. It was gathered in a folder and then compressed into a zip file. The file was uploaded to Google Drive for training and testing purposes.

The following section contains information about training outcomes. We use graphs and tables to display results for mAP, F1-Score, precision, recall, and IOU metrics. The total number of training iterations is 8000. After 1000 iterations, we displayed the values of all parameters. The proposed method outperforms existing state-of-the-art methods in terms of accuracy and universality. It has a wide range of applications in computer-assisted tooth treatment and diagnosis. Following training, the trained model YOLOv3 was tested on test images and achieved 99.33%.

### 5.1. Performance Evaluation

Criteria regression models are evaluated based on the mean average percentage. Precision, recall, accuracy, and F1 score are used to evaluate classification model performance. The object detection model’s goal is to classify and localise an object in an image. In our case, we must estimate the classification and localization performance. We will evaluate our model using the following parameters: mAP, recall, precision, F1- Score, and IOU.

**Training mAP (mean average precision**): “Taking the mean AP of all classes or overall IOU thresholds” yields the mAP.
(1)$$mAP=\frac{1}{n}\sum_{k=1}^{k=n}APk$$**F1-Score**: The F1 score is calculated as given below. The different values by using different iterations are shown in Figure 6.
(2)$$F1-Score=\frac{2}{\frac{1}{recall}+\frac{1}{precision}}$$**Sensitivity/Recall:** The recall can be calculated as in Equation (3). The recall values of using different iterations are shown in Figure 6.
(3)$$Recall=\frac{TP}{TP+FN}$$**IOU (intersection over union)**: IOU is used to assess the accuracy of object detectors on a given dataset. It computes the intersection of two bounding boxes’ union. The actual and predicted bounding boxes are:
(4)$$IOU=\frac{Area\;Of\;Overlap}{Area\;Of\;Unio}$$**Precision:** The precision metric displays “the positive class’s accuracy.” It assesses the likelihood that the positive class’s prediction is correct.”
(5)$$Precision=\frac{TP}{TP+FP}\;\;$$

Results of all the training parameters such as F1-score, recall, IOU, precision, and map on the total of 8000 iterations are given in Table 5.

### 5.2. Testing Results

In this section, we used weight files for testing our dataset. We have a total of 8 weight files that were generated during the training of the dataset every 1000 iterations.

**Testing mAP (mean average precision)****:** the mAP is calculated by “averaging the AP of all classes or overall IOU thresholds”**Testing F1-****Score**: the different values of the F1-score of testing using different iterations are shown in Figure 7.**Testing****Recall**: the different values of testing recall by using different iterations are shown in Figure 7.**Testing Intersection over Union****(IOU**): IOU values for testing using different iterations are shown in Figure 7.**Testing Precision**: testing precision values using different iterations are shown in Figure 7.

Results of all the testing parameters such as F1-score, recall, IOU, precision and map on the total of 8000 iterations are summarized in the following Table 6.

Output Images using the best weight obtained for all teeth diseases are shown in Figure 8.

We augmented the data with image rotation and shear range, among other things, to reduce the overtraining effect. The average classification accuracy when using augmented training data was 99.33%. Compared to the result without data augmentation, data augmentation showed a noticeable improvement in classification accuracy. This suggests that expanding the dataset will result in further improvements. Unlike the previous methods, the proposed method achieves high classification accuracy. The comparative study of the proposed model with recent DL models is shown in Table 7.

## 6. Conclusions

The most common dental issues from poor oral health care practices are cavities, root canals, dental crowns, and BDR. A dentist can detect potential dental problems by inspecting and gently moving the teeth. Automatic classification of dental conditions based on panoramic X-ray images can assist doctors in making accurate diagnoses. Panoramic dental radiographs are used to detect such tooth problems. To address poor efficiency, the complexity of the experiential operation, and high level of user intervention in existing methods of tooth problem detection, we propose a novel approach based on the deep learning model YOLOv3 for detecting and classifying the four most common tooth problems, namely cavities, root canal, dental crowns, and broken-down root. The availability of labelled medical datasets is a significant challenge in many automated tooth problem detection and classification applications; dental disease datasets are no exception. In this study, we used deep learning to create an automated tool capable of identifying and classifying dental problems on dental panoramic X-ray images (OPG). We created a custom dataset of dental X-rays due to a lack of a dataset. The dataset consists of dental panoramic images from various clinics with dental problems such as cavities, root canals, BDR, and dental crowns, among others. The dataset comprises 70% training images and 30% testing images. The implemented solution was evaluated using several metrics, including intersection over union, precision, recall, and F1-score for generated bounding box detections. The proposed method outperforms existing state-of-the-art methods in terms of accuracy, and has a wide range of applications in computer-assisted tooth treatment and diagnosis. The trained model YOLOv3 was tested on test images after training and achieved an accuracy of 99.33%. In the future, the proposed methodology will be real-time for teeth abnormalities detection. One can create a larger data set containing more teeth disease classes. In addition, the latest version of YOLO or other deep learning and machine learning models would be applied for great results.

## Figures and Tables

**Figure 1 sensors-22-07370-f001:**
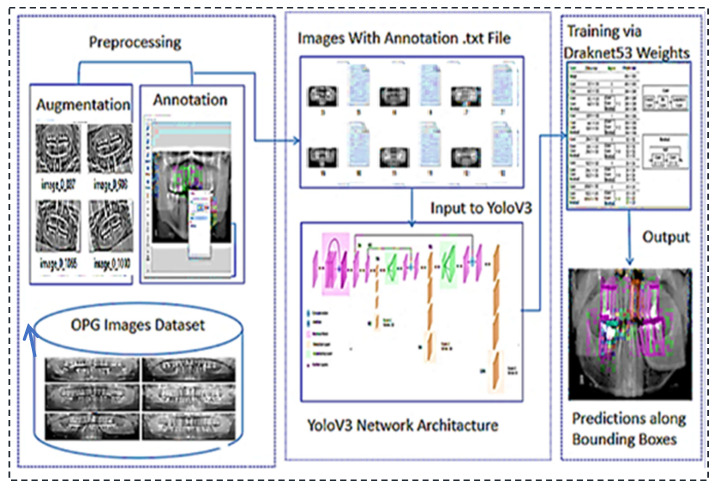
Proposed architecture’s workflow for the teeth diseases classification.

**Figure 2 sensors-22-07370-f002:**
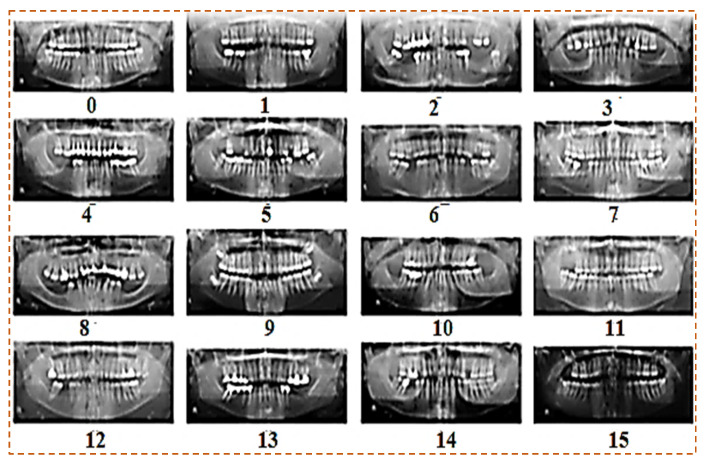
Custom dataset of panoramic X-rays.

**Figure 4 sensors-22-07370-f004:**
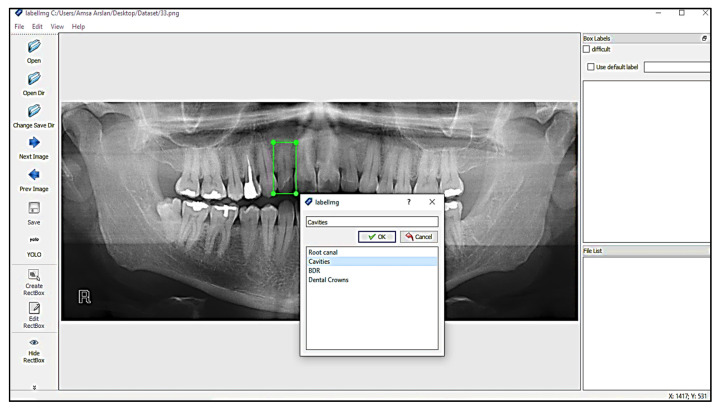
Labeling process with LabelImg.

**Figure 5 sensors-22-07370-f005:**
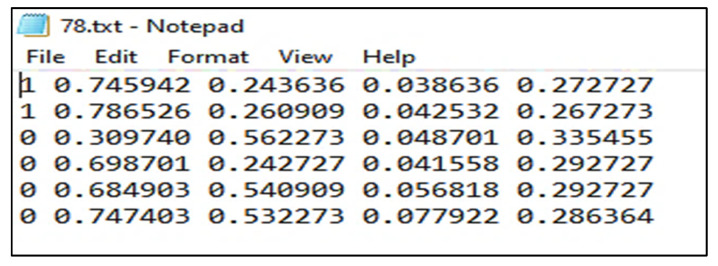
Annotation file in .txt format.

**Figure 6 sensors-22-07370-f006:**
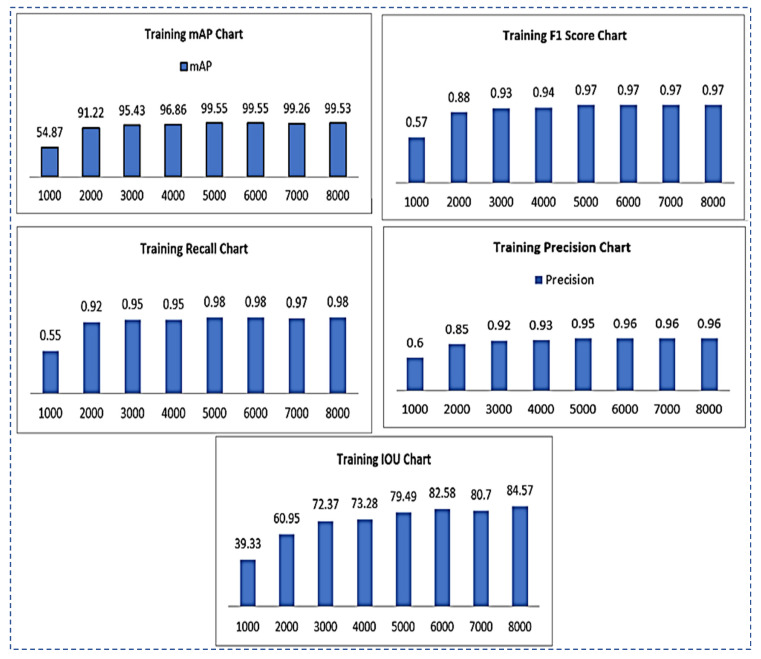
Training results chart representation for mAP, F1-Score, recall, precision, IOU.

**Figure 7 sensors-22-07370-f007:**
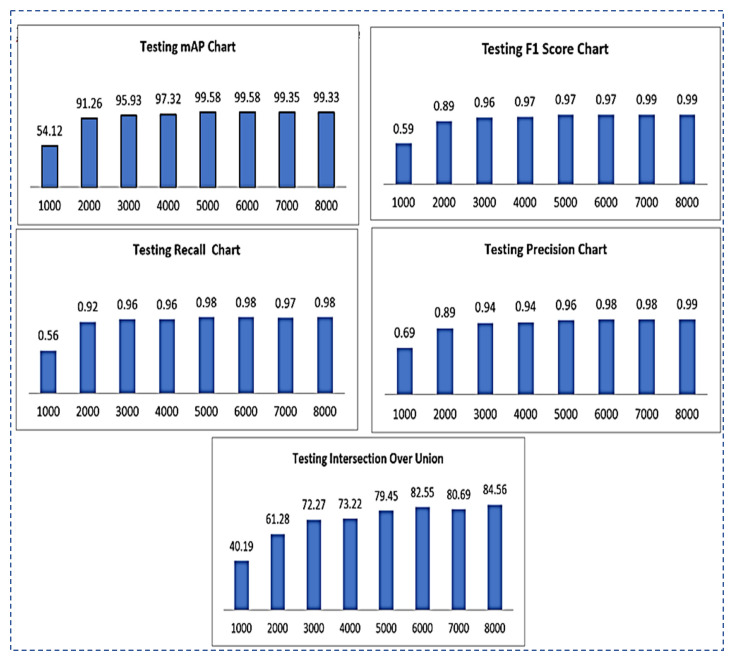
Testing results chart representation for mAP, F1-Score, recall, precision, IOU.

**Figure 8 sensors-22-07370-f008:**
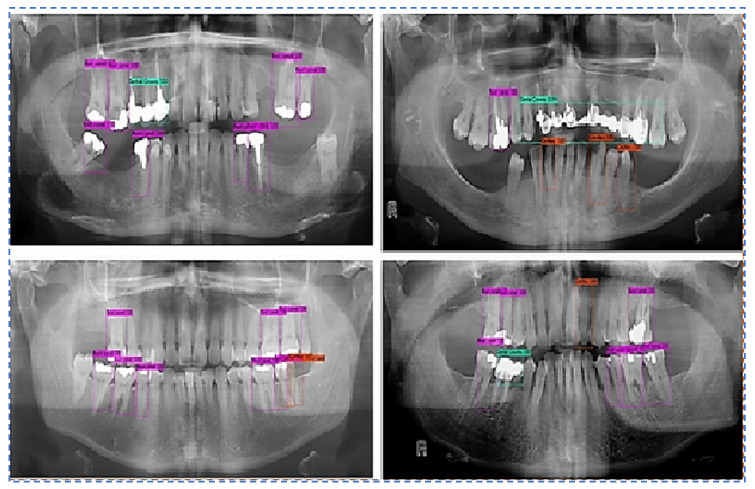
Predictions along bounding boxes using the best weight.

**Table 1 sensors-22-07370-t001:** Existing models for dental diseases.

Year	Modality	Dataset Size	Model	Accuracy	Authors
2021	OPG	100	ResNet	93.2%	Chisako Muramatsu et al.
2021	OPG	708	CNN	88.89%	Jun-Young Cha et al.
2021	OPG	420	CNN	90.36%	Liu et al.
2021	OPG	300	CNN	82.7%	Byung Su et al.
2020	OPG	206	CNN & RCNN	90%	Hassan Aqeel Khan et al.
2020	OPG	340	CNN	93%	Chang et al.
2020	OPG	83	SVM	93.6%	Abdalla-Aslan et al.
2020	OPG	100	CNN	81%	Thanathorn Won et al.
2020	OPG	680	CNN & VGG16	84%	Lee et al.
2019	OPG	353	CNN	81%	Krois et al.
2020	OPG	300	CNN	93%	Fukuda et al.
2019	OPG	85	Deep Feedforward CNN model	81%	Krois et al.
2019	OPG	200	CNN	87%	Bouchahma et al.

**Table 2 sensors-22-07370-t002:** Structure of the Darknet 53.

Layers	Filter Size		Repeat	Output Size
Convolutional	32	3 × 3/1	1	416 × 416
Convolutional	64	3 × 3/2	1	208 × 208
Convolutional Convolutional Residual	32 64	1 × 1/1 3 × 3/1	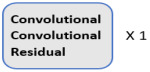	208 × 208 208 × 208 208 × 208
Convolutional	128	3 × 3/2	1	104 × 104
Convolutional Convolutional Residual	64 128	1 × 1/1 3 × 3/1	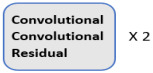	104 × 104 104 × 104 104 × 104
Convolutional	256	3 × 3/2		52 × 52
Convolutional Convolutional Residual	128 256	1 × 1/1 3 × 3/1	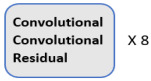	52 × 52 52 × 52 52 × 52
Convolutional	512	3 × 3/2		26 × 26
Convolutional Convolutional Residual	256 512	1 × 1/1 3 × 3/1	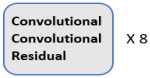	26 × 26 26 × 26 26 × 26
Convolutional	1024	3 × 3/2	1	13 × 13
Convolutional Convolutional Residual	512 1024	1 × 1/1 3 × 3/1	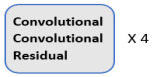	13 × 13 13 × 13 13 × 13

**Table 3 sensors-22-07370-t003:** Parameters of the custom configuration file.

Parameters	Values
Batch	64
Subdivisions	120
Width	416
Height	416
Channel	3
Learning rate	0.001
Max batches	80,000
Steps	7800, 8200
Classes	4
Filters	27

**Table 4 sensors-22-07370-t004:** Google Collab Parameters.

Parameters	Values
CUDA	10,010
cuDNN	7.6.5
GPU	NVIDIA Tesla K80 GPU
RAM	12 GB
DISK Space	68 GB

**Table 5 sensors-22-07370-t005:** Training results of F1-score, recall, IOU, precision, mAP.

Iteration	F1 Score	Recall	IOU	Precision	mAP
**1000**	0.57	0.55	39.33%	0.6	54.87%
**2000**	0.88	0.92	60.95%	0.85	91.22%
**3000**	0.93	0.95	72.37%	0.92	95.43%
**4000**	0.94	0.95	73.28%	0.93	96.86%
**5000**	0.97	0.98	79.49%	0.95	99.55%
**6000**	0.97	0.98	82.58%	0.96	99.55%
**7000**	0.97	0.97	80.7%	0.96	99.26%
**8000**	0.97	0.98	84.57%	0.96	99.53%

**Table 6 sensors-22-07370-t006:** Testing results of F1-score, recall, IOU, precision, mAP.

Iteration	F1 Score	Recall	IOU	Precision	mAP
1000	0.59	0.56	40.19%	0.69	54.12%
2000	0.89	0.92	61.28%	0.89	91.26%
3000	0.96	0.96	72.27%	0.94	95.93%
4000	0.97	0.96	73.22%	0.94	97.32%
5000	0.97	0.98	79.45%	0.96	99.58%
6000	0.97	0.98	82.55%	0.98	99.58%
7000	0.99	0.97	80.69%	0.98	99.35%
8000	0.99	0.98	84.56%	0.99	99.33%

**Table 7 sensors-22-07370-t007:** Comparative study of the proposed model with recent DL models.

Year	Dataset Size	Model	Accuracy	Authors
2019 [14]	800	Deep neural Transfer Network (DeNTNet)	Accuracy: 0.69 and F1 score: −0.75	Kim et al.
2018 [18]	800	Label tree with cascade network structure using CNN	F-score: 0.959, Precision: 0.958, Recall: 0.961	Zhang et al.
2020 [30]	105	By using the Back-propagation neural network	Accuracy: 0.971, ROC: 0.987, PRC: −0.987 learning rate value: 0.4, Momentum value: 0.2, FPR: 0.028	Geetha, Aprameya & Hinduja
2019 [24]	300	DetectNet with DIGITS	Precision: 0.93, Recall: 0.75, F-measure: 0.83	Fukuda et al.
2020 [33]	100	CNN (Resnet 50)	Sensitivity: 0.964, Average accuracy: 0.872, (multisided models): 0.932	Muramatsu et al.
2022 [23]	846	Mask R-CNN model	F1 score: 0.875, Precision: 0.858, Recall: 0.893, Mean ‘IoU’: 0.877	Lee et al.
2020 [24]	83	SVM	Accuracy: 93.6%	Abdalla-Aslan
2019 [26]	353	CNN	Accuracy: 81%	Krois et al.
2020 [27]	340	CNN	Accuracy: 93%	Chang et al.
2021 [28]	708	CNN	Accuracy: 88.89%	Jun-Young et al.
2021 [29]	300	CNN	Accuracy: 82.7%	Byung Su et al.
**Proposed Method**	**1200**	**YOLOv3**	**F1 score: 0.99, Recall: 0.98, Iou: 84.56%, Precision: 0.99, mAP: 99.33%**	**Proposed Work**

## Data Availability

The data could be available on request.

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
