# Peer review of "Deep Learning Models for Classification of Dental Diseases Using Orthopantomography X-ray OPG Images"

_sensors, 2022, doi:10.3390/s22197370_

Round 1
Reviewer 1 Report
Find the attached file

Reviewer 2 Report
The work titled ‘’ proposed a method for detecting and classifying the four most common tooth problems: cavities, root canals, dental crowns, and broken down root canals based on the neural learning model YoloV3. This works aims to create an accurate classification of these dental problems. The work is interesting for researchers however it can be published after addressing some concerns/issues.
1. Abstract is too long. It can be shorten.
2. Clearly mention the contributions in introduction section.
3. Are annotations validated by dental specialist?
4. Architecture provided in Figure 6 need to presents in tabular required.
5. What is the role of residual layer in Darknet? Discuss.
6. Table 2. Parameters of custom configuration file. Are these parameters optimized?
7. Table 3: Google Collab Parameters. Are these parameters or configuration?
8. The LR section need to be enhanced. The authors should include and discuss more related work in other application for a broader vision. Further, authors should add more related work as provided below.
· Tiwari, S., & Jain, A. (2021). Convolutional capsule network for COVID‐19 detection using radiography images. International Journal of Imaging Systems and Technology, 31(2), 525-539.
· Zhang, X., Liang, Y., Li, W., Liu, C., Gu, D., Sun, W., & Miao, L. (2022). Development and evaluation of deep learning for screening dental caries from oral photographs. Oral diseases, 28(1), 173-181.
99. Using k-fold cross-validation is more accurate. It is better to report your results using k-fold cross-validation.
Round 2
Reviewer 1 Report
Now the reviewer accepts this manuscript for the Sensors journal because it was significantly improved the quality compared to the previous version.
